# People Living with HIV and AIDS: Experiences towards Antiretroviral Therapy, Paradigm Changes, Coping, Stigma, and Discrimination—A Grounded Theory Study

**DOI:** 10.3390/ijerph20043000

**Published:** 2023-02-09

**Authors:** Helmut Beichler, Ruth Kutalek, Thomas E. Dorner

**Affiliations:** 1Nursing School, General Hospital, Medical University Vienna, 1090 Vienna, Austria; 2Department of Social and Preventive Medicine, Centre for Public Health, Medical University of Vienna, 1090 Vienna, Austria; 3Academy for Ageing Research, Haus der Barmherzigkeit, 1160 Vienna, Austria; 4Department of Social and Preventive Medicine, Centre for Public Health, Medical University of Vienna, Kinderspitalgasse 15/1, 1090 Vienna, Austria

**Keywords:** ART, adherence, coping, stigmatization, psychosocial aspects, health care professionals

## Abstract

Background: The experiences in coping with HIV/AIDS from people living with HIV (PLWH) in Austria, Munich, and Berlin regarding adherence, antiretroviral therapy (ART), stigmatization, and discrimination were the main focus of this study. Therapy adherence is the cornerstone for PLWH to reduce disease progression and increase life expectancy combined with a high quality of life. The experience of stigmatization and discrimination in different life situations and settings is still experienced today. Aims: We aimed to examine the subjective perspective of PLWH concerning living with, coping with, and managing HIV/AIDS in daily life. Methods: Grounded Theory Methodology (GTM) was used. Data collection was conducted with semi-structured face-to-face interviews with 25 participants. Data analysis was performed in three steps, open, axial, and selective coding. Results: Five categories emerged, which included the following: (1) fast coping with diagnosis, (2) psychosocial burden due to HIV, (3) ART as a necessity, (4) building trust in HIV disclosure, (5) stigmatization and discrimination are still existing. Conclusion: In conclusion, it can be said that it is not the disease itself that causes the greatest stress, but the process of coping with the diagnosis. Therapy, as well as lifelong adherence, is hardly worth mentioning today. Much more significant is currently still the burden of discrimination and stigmatization.

## 1. Introduction

In the last 40 years since the outbreak of the HIV pandemic, more than 30 million people have died of AIDS [1] Human immunodeficiency virus (HIV) and acquired immune deficiency syndrome (AIDS) continue to spread worldwide. The efficacy of ART with modern drug combinations for the treatment of HIV/AIDS allows many people, despite their chronic disease, to have a life expectancy comparable to that of people without the infection [2]. 

The association among a high CD4 cell count, undetectable viral load, reduced comorbidities, and high quality of life has been confirmed [3]. Improving socioeconomic disparities, clinical factors, holistic treatment management, and health policy strategies for prevention, education, and also reducing stigma and discrimination influence treatment outcomes and improve quality of life [4]. Overall, this has led to a paradigm change for PLWH both in addressing the diagnosis and in the daily management to cope with the chronic disease. Particularly in highly industrialized countries, HIV can now be integrated and managed particularly well in daily life. The modernization of diagnostics and early initiation of antiretroviral therapy make optimal disease management possible, accompanied by seamless care of the treatment team [5]. Women living with HIV occupy a special position due to the prevalence of transmission, social positioning, vulnerability, need for pregnancy, risk of premature birth, and earlier onset of menopause but also violence against women [6].

Planning for the personal future, due to the reduction of mortality especially in industrialized countries with access to ART, is possible today [7]. 

In the EU countries, approximately 25,000 people were diagnosed with HIV in 2021.

PLWH treated according to international recommendations can reach an almost normal life expectancy with good quality of life [8]. A high level of adherence is necessary for successful treatment with ART [9]. This corresponds with agreed recommendations, shared decision-making, patients’ perception, and experiences from the Health-Care Professionals (HCPs) as cornerstones for ART adherence [10]. Despite the decreasing fear of AIDS and the paradigm change by the reduction of old clichés, PLWH are still confronted with discrimination and stigmatization in different areas of life [11]. Goffman (1979) [12]. describes stigma as physical signs burnt into the bodies of social outcasts in ancient Greece to reveal their social status. Stigmatization is discrediting, and discrimination results from it [13]. Furthermore, stigma has an impact on the health and well-being of individuals, particularly as it induces high rates of psycho-sociological stress [14]. PLWH especially in combination with homosexuality, bisexuality, high promiscuity, and intravenous drug use are groups that are particularly discriminated.

### Aims

This grounded theory study aimed to examine the subjective perspective of PLWH concerning living with, coping with, and managing HIV/AIDS. We also aimed to describe an up-to-date picture of PLWH including their personal views on living with HIV/AIDS in everyday life, at work, and in society in Austria and Germany.

## 2. Materials and Methods

Grounded Theory Methodology (GTM) was used to describe and conceptualize people’s views, actions, and life experiences [15]. 

### 2.1. Recruitment and Participation

People participating in the study were recruited via physicians and through civil society organizations. Participants were made aware of the study via flyers and were directly approached and referred by HCP from January to September 2022.

### 2.2. Ethical Consideration

Ethics approval was provided by the Ethics Committee of the Medical University of Vienna (EK no. 1610/2019). Before the interviews, participants were provided with information on the study with an opportunity to seek additional information or clarification. Participants gave their written informed consent that included that all data were anonymous with code names assigned to all participants.

### 2.3. Data Collection and Analysis

All semi-structured face-to-face interviews were conducted by the first author. The thematic construction of the interview guide (Table 1) was based on the phenomena formulated in the research questions and the current research literature. Interviews were digitally recorded and transcribed verbatim. Interviews lasted 45–90 min. Interviews were mostly performed via video-call.

Based on Strauss and Corbin’s (1996) suggestions for an inductive approach to interviewing questions, each interview started with an open-ended question asking participants “how are you doing right now?”

Data analysis was performed in three steps, open, axial, and selective coding. The analysis was supported by MAXQDA 2022 version 22.2.0 software. The theoretical sampling gives the researcher a continuous direction within the data collection process.

## 3. Results

### 3.1. Participants

A total of 25 interviewed participants (*n* = 25, 10 women, and 15 men, aged 23 to 65 years) participated in the study (Table 2). All PLWH had been diagnosed with HIV of varying durations (mean duration of HIV infection in years = 18.0 years, SD 13.1). The time of diagnosis varied from 1983 to 2021. Most male participants are homosexual and have sex exclusively with men. One of them was heterosexual and got infected during male contact. Some of the men were infected by their partners in existing partnerships. Some men were infected with HIV at the beginning of the HIV pandemic in the 1980s where people did not yet know about HIV. All interviewed women were infected by their HIV-positive partners in existing partnerships. Additionally, it should be noted that most of the participants never talked about HIV with their partners.

The analysis of the qualitative data generated five core categories: (1) coping with the diagnosis through a paradigm shift, (2) psychosocial burden of HIV, (3) ART as a need, (4) trust as a requirement in HIV disclosure, (5) stigmatization and discrimination. Furthermore, subcategories were formed for each core category (Figure 1).

### 3.2. Coding Paradigm

The central coding paradigm (Figure 2) consists of the phenomenon of “fast coping with diagnosis is possible due to paradigm change”, the causal condition “psychosocial burden caused by the HIV diagnosis”, the intervening conditions “relationship to Health Care Professionals, Shared decision making, health literacy”, the contextual conditions “stigmatizing and discrimination”, as well as the main strategy “immediate start of the antiretroviral therapy”, and the consequences if the management of the overall situation is successful and the need for disclosure in different settings and situations.

The central phenomenon is influenced by stigma and discrimination as the temporal, local conditions, social and cultural environment, and individual biography of PLWH and with intervening conditions as the relationship with HCPs, the coping strategy (anonymity, discretion) to avoid stigma and discrimination, adherence as a self-imposed and external commitment, health literacy, and shared decision making.

### 3.3. Central Phenomenon—Fast Coping with Diagnosis through a Paradigm Change

The central phenomenon as a successful way to deal with challenges is characterized by the rapid development of the first coping strategies to overcome the shock and quickly regain the ability to act in daily life. The diagnosis is directly connected with the beginning of the therapy. This gave most of the participants a high degree of security together with health care professionals.

The group of participants interviewed was a very heterogeneous group. One thing remained the same for all persons—the diagnosis always came unexpectedly with the person being unprepared and was surprisingly associated with psychosocial crisis and shock.

The paradigm shifts in particular were experienced by most of the participants interviewed as an advantage in coping with HIV infection, due to the possibility of antiretroviral therapy immediately after diagnosis. In particular, the ease with which the tablets can be taken means that the HIV diagnosis can be quickly integrated into everyday private and professional life without any particular challenges.

The communication of the positive test result by professionals was characterized by shock, psychosocial crisis, stress, surprise, excessive demands, inability to act, loss of control, and above all, the loss of self-confidence for the participants interviewed. HIV represents the worst thing that has to be accepted, less the coping with the diagnosis in everyday life.

“*The diagnosis really crushed me, it crushed me. But not for a long time. From being a woman who is ready to fight and wants to get through somehow…that’s when I was just crushed, that’s when I really ran out of steam at one point*”.(I. 16 10–10)

“*When I got the news, it pulled the rug out from under me. I walked through the streets like I was wrapped in absorbent cotton. A world collapsed for me, I was in panic*”.(I. 22, 13–13)

PLWH described coping as an actively initiated process of learning and confronting how to deal with the (new) situation, including emerging future problems. The paradigm shift has occurred due to the reduction of stereotypes, as well as the improved treatability with modern ART. This led to rapid coping with the currently shocking and overwhelming diagnosis.

“*I intentionally said to myself, ‘You’ve got to get a handle on this now.’ I actively dealt with myself and the new situation. That hurt, but I knew I wasn’t lost, and I didn’t have to die, that’s what my doctor told me right away, that relieved me a lot. But you have to do it actively yourself. Just sitting around and waiting doesn’t work, I realized that*”.(I. 16, 34–35)

Successful coping, usually with psychosocial support from professionals, leads to the ability to act and acceptance of the diagnosis. Participants also describe coping with the diagnosis as an opportunity for further development as a grown personality with new professional and private opportunities. Successful coping with the diagnosis results in a strengthened self-confident personality.

“*Today I am a consolidated and stable personality, good styling, I like myself, I am more confident today. Many react on the basis of old clichés and movies. Most of it took place in their own imagination. I’m glad I fought back in all the negative confrontations. Doing that made me more self-confident*”.(I. 16, 53–53)

The integration of HIV into everyday life is nevertheless described under the aspect that HIV remains as an integral part of life and can never be completely excluded through regular confrontation with the topic. PLWH also describe that the quality of life, although HIV indirectly determines life, is not affected by thoughts of HIV.

“*It’s difficult. It’s been in my life for the last 32 years and it’s just a theme. It’s part of my life. For me, quality of life is being healthy and being comfortable in my body*”.(I. 22, 21–21)

For most of the participants, the diagnosis is communicated to the future treatment team. A relationship develops between the physicians and nurses. This relationship grows through trust, especially through the regular three-monthly check-ups, which are an integral part of the treatment process. One woman specifically chose a physician who has a high level of psychosocial competence. Health literacy, with the need for continuous information and knowledge about individual blood results and check-ups, is also increasing among the participants. The opportunity to discuss treatment, changes and breaks in therapy with the physician, but also the interest in and significance of the individual laboratory parameters (especially the viral load and CD4 cell count) is experienced as a need.

“*Almost a bit like a relationship. Where you also sometimes argue but still totally appreciate each other and are glad to have the other person. I also just trust him. That’s what I’m so happy about, that I have such a doctor that I can trust. It’s such a personal relationship*”.(I. 9, 6–6)

“*I look for the doctors, nurses to have psychosocial competence. That’s a sign that you can talk about worries, fears, problems, that you also get to talk. That you are not immediately turned away in the office*”.(I. 13, 54–54)

In a special way, all participants emphasize this paradigm shift, which is mainly due to ART. Those who became infected with HIV after 1996 have benefited the most, because the threat of the disease has been reduced to a minimum. This makes rapid management possible, and HIV no longer dominates everyday life due to the absence of side effects.

“*There has been a paradigm shift between the knowledge of HIV from before and to what you get about HIV in terms of knowledge. HIV is no longer a problem today. The threat that you have a deadly disease with no chance is gone. It’s really just one pill a day, no more, no less. The importance of adherence has to do with viral load, that guarantees I can live to be old for a long time*”.(I. 17, 15–15)

### 3.4. The Psychosocial Burden Caused by HIV

Psychosocial burden caused by HIV is characterized by shock, overwhelm, surprise, coping as an actively initiated process, relationship with the HCP, threat potential minimized, and personal growth. The psychosocial burden dominates with the loss of self-confidence, shame, feeling of guilt, social withdrawal, loss of joy of life, self-isolation, social isolation, loneliness, and depression, and some participants reported suicidal thoughts.

Some of the PLWH said that they could not focus on anything else during this phase. Women in particular described psychosocial stress, social withdrawal, self-isolation, loneliness, loss of joy in life, feelings of shame and guilt, and suicidal thoughts. Men did not report feelings of shame and guilt during this phase.

“*What HIV does, it gnaws away at your self-esteem. You feel inferior. You feel like a second-class person, and you radiate that. It depends on how you communicate yourself about HIV and that’s how it comes across to the other person*”.(I. 21, 27–27)

“*The main aspect for me was really the topics of sexuality, shame and guilt. Coming to terms with the diagnosis, being able to accept it and this topic of shame and guilt and dealing with one’s own sexuality. That was very intense for me. I was in this shame and guilt theme. It’s not easy to talk about it with a subject. Talking with another woman about HIV*”.(I. 16, 53–53)

One man is crushed by the diagnosis to this day (3 years) and has been unable to find coping strategies to integrate HIV into everyday life. Suicidal thoughts regularly dominate the day.

“*The flame that you have as a zest for life, that you want to go out, meet new people, that you might meet someone again on Tinder…but I don’t even want that anymore. It’s about socializing, which I avoid*”.(I. 3, 8–8)

“*I’m already also always struggling because I realize that I also keep falling into a hole and I’m not satisfied. That with my 56 years I’m still sitting in a rented apartment. I have no children and no partner. These are sometimes things that pull me down. But then I work on it. I can often no longer judge what comes from where. You don’t know if you have something physical or psychological if it comes from the medication or from HIV. I can’t tell anymore. Or is it the whole normal madness that occurs*”.(I. 22, 22–22)

### 3.5. Antiretroviral Therapy as a Need

Regarding the ART, the focus is on an undetectable viral load. Adherence concerning ART is self-imposed, with external commitment, the development of consequence, continuity, and health literacy as a need for shared decision making.

The prescription of therapy is closely related to relationship building with the prospective treatment team. The immediate and prompt start of ART, especially among participants who contracted it between 2010 and 2022, is a great need, with the feeling that “something is being done against the virus”, and this suggests safety and protection of the immune system.

“*For me it was important, this has to start right now. I needed this feeling; something has to be done against this virus. I was also very worried about my immune system. When I think about how many pills people take for any disease… I have to take one pill. If that is the solution, then the solution is very simple*”.(I. 17, 12–12)

The viral load parameter has become the decisive health parameter and stands for health, well-being, and quality of life, but also for being able to live with the chronic disease HIV.

“*It is important to me that I know every 3 months that the viral load fits anyway and that I am not infectious. That’s why I’m very consistent with taking ART. It is so routine and so automatic in my life to swallow these pills that I even have to be careful not to take them twice*”.(I. 19, 23–23)

Adherence is described by all 25 participants as manageable, with no burdens or restrictions on the quality of life. The simple drug dosage (single dose), frequency of intake, and the good tolerability of the ART enable rapid integration of ART in daily life without adaptation.

“*I started ART right away. It worked great, right from the first dose. Adherence is not a problem for me, it works. I write it down every day, it’s ritualized*”.(I. 24, 11–11)

“*This is luxury these days. I take one tablet a day. That’s already an automatism. I never forget, it’s already in the flesh. I used to have to take a handful of tablets. Lots of side effects, lots of extra tablets to make the side effects bearable*”.(I. 22, 25–27)

The motivation for adherent behavior also relates to compliance with the control examinations (three-monthly viral load determination, CD4 cell count), as well as ensuring a sufficient supply of pills. Furthermore, the importance of adherence as a commitment to self and society was described. The care paradigm Undetectable is Untransmittable (U=U) developed by The Joint United Nations Programme on HIV/AIDS (UNAIDS) becomes the metaparadigm in the lives of all patients interviewed.

Even if the experts guarantee that the virus will not be transmitted, most participants are still very careful and describe fear of infecting someone. Couples with a serodiscordant HIV status will not practice unprotected intercourse.

“*I know I’m not infectious, but I don’t trust it 100 percent. Who knows, then I have more viruses in the body, and I infect someone, I could never agree with my conscience. We do only protected sexual intercourse*”.(I. 3, 12–13)

Moreover, the idea to prevent the further global spread of HIV, as well as the protection of people, is in the foreground of the participants.

“*The prevention idea is already there. Take the one tablet, keep the same timing, then that’s your lifesaver*”.(I. 17, 15–17)

“*It’s important to me that nothing can happen, that I can’t infect anyone. That’s what’s in my head, just don’t infect anyone*”.(I. 20, 13–13)

Health literacy with the need to take responsibility for the disease and to be involved in all treatment decisions (shared decision-making) is particularly important to most of the participants.

They want to take responsibility for the disease and have a say in all treatment decisions. They want to know what is happening to their body. The importance of continuous information about the individual blood results and the results of the control examinations are important to know. The psychosocial competencies of the HCP were emphasized by the participants.

“*Having a say is very important to me. I want to have a say, it’s my body and my HIV. It does everything I want it to do, if I want any minerals or anything checked out. I am very active myself, what I want to have examined. I want to know what’s going on inside me, after all*”.(I. 24, 24–25)

“*It’s totally important to me. I’m very interested in it. I always look at the values very carefully. I read up a lot and also research corresponding studies and read that. I find out about diseases immediately. I want to know everything exactly. I want to know what is in store for me. I also want to know about the effects of my medications*”.(I. 14, 70–70)

### 3.6. Trust as a Requirement for HIV Disclosure

HIV disclosure as a need has become visible through the issues strengthened, self-confidence, integration of HIV into everyday life, necessity for sexuality, intimacy, partnership, through outing HIV gets space in life, trust as a requirement, and not to be reduced to HIV.

With coping and integrating HIV into everyday life, comes the feeling for partnership, friendship, and sexuality. Especially women said that they talked about the HIV diagnosis when they met a new potential partner for a relationship. Trust had to be built first so that the diagnosis could then be shared like a thoughtful process.

In addition, HIV outing is carefully analyzed and weighed with advantages and disadvantages. This has an influence on the process of coming out.

“*It depends on do I say that to a neutral person, or do I say that to a man I want to go to have sex with…. My current boyfriend knows from the first night from my HIV-infection because I knew it was going to be more than a one-night stand. You can’t miss the moment to disclose the diagnose. You can’t have sex with someone before you talked about HIV. It has to be before. Otherwise, the trust is broken*”.(I. 21, 27–27)

“*HIV is a wall that has to be overcome. I’m not waiting for the right moment. If you don’t say that and have with sex, maybe that’s a cut you’ll never be in a relationship with then. Because of the basis of trust. That’s where you put obstacles in your own way*”.(I. 13, 64–64)

Once the diagnosis has been processed and ART has become ritualized in everyday life, the need for a self-confident and self-determined life in society was described by the participants. Disclosure of the diagnosis was considered when it was absolutely necessary to share the diagnosis, for example, when visiting a doctor for a specific treatment (surgical procedure) and it was important to talk about the HIV infection.

“*I think very specifically about who I tell about my HIV infection. I always ask myself whether this information is absolutely necessary now. And I ask myself whether I myself have a benefit, e.g., if I am ill and it is necessary that I tell my doctor, then I also do that. This is also about my own health and safety*”.(I 6, 13–15)

Some participants told us that they also do not want to be reduced to HIV with stereotypes.

“*The most important thing for me is that I don’t want to be reduced to HIV. To be reduced to something that is not relevant in a friendship or in a family. HIV is still tainted with old stereotypes that are no longer relevant today*”.(I. 19, 35–38)

“*Of course, you are special. So, I already like to be the center of attention (laughs), but not as far as HIV is concerned, because it just has a negative touch at the end of the day and I don’t want to be reduced to that, I don’t want to be reduced to a disease or now ultimately an infection. But rather I want to be perceived as a human being at the end of the day*”.(I. 9, 110–110)

Especially women talked about gender-specific aspects and described their insecurities. Women described that during pregnancy, as well as birth, the information about HIV is necessary and the diagnosis was disclosed to the doctors. The certainty of not infecting the unborn child if the viral load is not detectable is not always fully given by the participants.

Women describe themselves as more vulnerable than men and that this vulnerability also has a negative impact on their self-esteem. Ideals of beauty, especially when there is a body image disorder due to a change in the figure (lipodystrophy syndrome), are in the foreground. Fear of the disease becoming visible to society and an unwanted outing of the person is a concern for PLWH.

“*I also suffer from having this fat distribution disorder. I got a big uneven breast. I suffered a lot from that. That’s when I had a breast reduction. It has become better. Of course, the beauty also suffers, and the disease becomes visible*”.(I. 18, 55–55)

### 3.7. Stigmatization and Discrimination

Stigmatization and discrimination are complex and characterized by discreditation above all in the workplace, stigmatization through stereotypes, self-stigmatization, discrimination by HCPs, the coping strategy anonymity and discretion, and freedom through being HIV-restricted (Figure 3). Stigmatization and discrimination have a significant influence on how the diagnosis is dealt with externally in society and can be an obstacle to adherence to ART. In addition, the overall burden of the situation is intensified. In the context of outing the diagnosis, the risk of discrimination increases.

The causes of stigma are rooted in the fear of HIV. Especially the older participants, those who were some of the first to become infected with HIV in the 1980s, were filled with misconceptions about HIV as well. Many of our perceptions about HIV stem from the HIV images of the early 1980s. There are still misconceptions about HIV transmission and what it means to be infected.

Two participants reported that they suffered from loss of appetite and wasting syndrome for a period of time due to HIV. The result was high weight loss with weakness and fatigue. Long periods of inability to work were the result. From the employers’ point of view, the reason to question the situation was with the result that the pressure increased by hiding the HIV infection.

“*I have had no appetite for a long time and have lost a lot of weight. I was very tired, and I felt very bad. I was listless and my circulation did not work. It was hard for me to get through the workday because of these fatigue attacks. My boss often asked me what was wrong with me, and I always had to make excuses and lie. But it was obvious to everyone that I was seriously ill. At some point I couldn’t take it anymore and told them I was diagnosed with HIV. After that, I was no longer acceptable at work because of the diagnosis*”.(I. 5, 34–35)

One woman reported that the employer was able to find out about the HIV infection in a roundabout way. In this case, termination also followed. The level of material threat has increased for this reason.

The lack of information in society combined with the old stereotypes and perceptions causes people to be afraid of contracting HIV. In addition, people think that HIV infection is reserved for a certain group. This in turn leads to negative values and value judgments about PLWH.

“*If more social acceptance was there, where you could talk about HIV exactly o, like you could talk about a herniated disc, for example, you could talk about it more easily and much more often. Not always just this stigma, you can’t talk about it*”.(I. 16, 68–69)

Discrimination by HCPs, including loss of job, was experienced by some participants. PLWH described HCPs as refusing to provide services. Women respondents in particular described that discrimination still occurs against PLWH by refusing an appointment by the gynecologist.

“*I looked for a gynecologist for a long time, no one would take me. Now I finally found one, but I always get the appointment last thing in the evening…*”.(I. 23, 15–16)

Most of the participants interviewed indicated that discrimination was especially prevalent among dentists and dermatologists.

“*The Dentist turned me down. I was there I was in shock*”.(I. 22, 14–14)

“*When I told the dermatologist that I am HIV positive to protect her…..she insulted me and kicked me out of the office…*”.(I. 16, 13–13)

Especially in rural areas, PLWH are even more exposed to stigma and discrimination. Possible job loss results in material threats to livelihood, social isolation, loss of friendships and family, loneliness, and depression.

“*…then the virus got a point of attack… I lost a lot of weight, I got emaciated, people were pestering me at work, what’s wrong with me? A bad pressure built up in my head that I could no longer hold. And then when I came out, the bullying started, from my bosses. They said, you can’t continue this work anymore… I got worse and worse until I was fired*”.(I. 11, 16–16)

PLWH internalize the stigma and begin to develop a negative self-image. Feelings of being reduced to HIV develop. Participants described a fear of discrimination, as well as the uncertainty of being stigmatized and discriminated against by coming out.

“*Now I myself belong to this rabble… Yes, I don’t want to call myself an HIV hater, for me it was just hard to become one of them. Of course, that doesn’t sound nice, but that’s how I saw it and meanwhile I understand the people. But still there is a side with me where I think oh God, you are one of these freaks, yes exactly*”.(I. 3, 18–18)

“*Stigmatization is still as bad in certain strata of society as it was 20 years ago. It’s still a dirty disease. It still has that connotation. But people don’t know that HIV-positive people are no longer contagious. That has not been conveyed in the media*”.(I. 21, 22–22)

This gives rise to increased vigilance for most, as well as anonymity and discretion as a strategy. Nevertheless, outing the diagnosis is sometimes necessary to gain support from professionals in the health care system. This simultaneously risks exposure to stigma and discrimination. The patients interviewed argued for more social acceptance and equal treatment to live openly with their disease.

“*It’s fundamentally embarrassing to me, it’s so very afflicted. It’s so badly afflicted. With homosexuality, drugs, that’s enough, that’s already 90 percent of people what goes through their heads when they think of HIV, forget it, never. I would advise everyone not to say this to anyone, especially at work. Because there are people who are like I used to be, who don’t want to have anything to do with people like that. It’s too negative, I had to learn to see it from a different perspective, that I’m not a monster*”.(I. 3, 46–46)

## 4. Discussion

This grounded theory study aimed to explore the experiences of how PLWH in Austria, Berlin, and Munich describe their everyday life and what challenges in coping and burdens have to be overcome. We also explored the experiences with ART, adherence, coping with the diagnosis, and the experiences about the relationship with their HCP. Furthermore, the investigation of experiences of stigma and discrimination was an additional aim. We identified five categories as crucial outcomes. The diagnosis is shocking and triggers a loss of self-confidence. Coping is challenging and intensely but briefly dominates daily life. Ferlatte et al. (2022) [16]. confirmed that suicidal ideation is particularly high among MSMs in the context of an HIV diagnosis. Our results are in line with another study that clearly showed that the diagnosis of HIV is still devastating news for PLWH, associated with a psychosocial stress response, shock, powerlessness, anxiety, and depression [17]. Interestingly, it has been found that processing shock and gaining agency take only a short time and the threat of HIV/AIDS is significantly reduced due to good treatability. The communication of the diagnosis and the prescription of ART simultaneously are state of the art. This is in line with DAIG (2022) [18], showing that the viral load is effectively suppressed below the detectability limit in the first two to four months. Knight et al. (2019) [19]. denied this finding and pointed out that the initiation of ART immediately after a diagnosis predominantly refers to countries where universal and unrestricted health care is available for PLWH. Areas of sub-Saharan Africa but also countries in Asia do not have these possibilities. The ritualized taking of pills, mustering high adherence, continuity, and integration of therapy into daily life are managed without challenges. This is in line with other studies that found that a viral load below the detectability threshold and CD4 cell count become important health determinants [20]. 

The relationship with the HCP, as well as the development of health literacy, increased self-confidence and provided reassurance [21]. Outing the HIV diagnosis is mostly a deliberate process based on trust with the other person. This is in line with Bickel et al. (2020) [22] who found that stigma and discrimination still occurred and were prevented primarily by anonymity and discretion.

Men differed from women in how they coped with the diagnosis. Men were more inclined to process the new situation through distraction, blame, displacement, and low growth than women. Women were ashamed but actively processed the diagnosis, got information about HIV, and tried to cope with the situation. Loss of self-esteem, feelings of shame and guilt, and self-stigma made women especially vulnerable during the diagnosis processing phase. These findings were confirmed by other studies, which describe that women coped with their diagnosis more constructively than men [23]. In addition, the study confirmed that men were less vulnerable and did not suffer as much loss of self-confidence as women as a result of the diagnosis [24]. The association between emotion-oriented coping strategies and social support with anxiety symptoms and depression has been confirmed [25]. Insecurity, the inability to act, hopelessness, sadness, stigma, and discrimination call for emotional support [26]. With effective ART, the early diagnosis and treatment of HIV are defining issues from an individual and public health perspective, in terms of prevention [27]. HCP relationships and resilience building are important pillars for ART adherence [28]. With the communication of the diagnosis, relationship building and trust between patients and HCPs is paramount [29]. HCPs pursue with the patient the goal of facts about HIV/AIDS, ART adherence and treatment success, and the importance of an undetected viral load in the context of sexuality Regarding complications, resistance, side effects, long-term toxicities, and comorbidities, PLWH are counseled individually [30]. Modula & Ramkumba (2018) [31] onfirm that more than 50 percent of PLWH suffer from mental health problems and recommend the implementation of mental health screening programs. Maintaining stable ART adherence is the goal for personal and global treatment success (UNAIDS 90-90-90 goals) [32]. The person’s lifestyle, internal doubts regarding treatment, rejection of the diagnosis, and the new reality can be barriers to ART success and must be evaluated in advance [33]. 

Side effects are only described in the initial phase of ART [34]. This was not confirmed by most of our interviewed participants; on the contrary, side effects were hardly mentioned.

ART’s occupational integration continued workability and a reduction in sick leave guarantee job retention and reduce material threats to livelihood [35]. Further, the appearance of good health mitigates HIV-related stigma and provides the respect to “live normally” [34].

HIV-induced stigma and discrimination have a high prevalence worldwide. The consequences of HIV-related stigma and discrimination are far-reaching and include delays in seeking HIV testing, denial of access to health care facilities, and non-disclosure of the HIV status [36]. PLWH suffer rejection, isolation, and emotional distress [37]. Not only patients but also HCPs, especially in areas with a high HIV prevalence, face discrimination in their work environment [38]. 

Stigma in health care settings is a barrier to ending the HIV epidemic [39]. Further, HIV-induced stigma has been shown to trigger psychological distress and low self-efficacy and negatively impact adherence [40]. PLWH interviewed in our study were unable to confirm negative effects on adherence from experienced stigma and discrimination. Moreover, AIDS stigma interacts with HIV prevention [41]. Internalized stigma, or self-stigma, occurs when a person takes the negative beliefs and stereotypes about PLWH and begins to apply them to themselves [42]. 

Discrimination is managed by PLWH by remaining anonymous and discreet in their daily lives. If HIV-induced stigma negatively impacts adherence, stigma management strategies become necessary [43].

Outing the diagnosis is a requirement of the PLWH interviewed in our study and at the same time, is directly related to the fear of stigma and discrimination, rejection, and disappointment. However, the association between voluntary and involuntary outings and medication adherence has been demonstrated [44]. Our study confirms that the disclosure of an HIV diagnosis only occurs under the condition of a high level of trust, in the relationship with the partner, friendship, or family. It was confirmed that HIV has little relevance from a PLWH perspective, and for this reason, anonymity and discretion are preferred [45]. Gabbidon et al. (2020) [46] escribe a decision-making and outing process with HIV self-disclosure. Studies confirmed relationship quality as a determinant of risk and resilience factors in the disclosure process of HIV infection. The HIV disclosure is also associated with the risk of emotional abuse of the person [47]. Also concerns about unintentional outings are high overall [48]. Bogart et al. (2008) [49]. describe that parents of HIV-positive relatives especially feel stigmatized and try to prevent outings.

Disclosing HIV to MSM is problematic in the family setting because homosexual sex practices are stereotyped and associated with HIV infection [50]. 

Additionally, it was confirmed that women, because of increased vulnerability, are more fearful of HIV disclosure, but more likely to choose to talk about HIV with their families and overall than men [51]. HIV disclosure is particularly relevant when sexual partners change frequently, especially when U=U cannot be guaranteed [52]. Also confirmed is that the concealment (outing is not an option) of HIV is higher among heterosexual men than women, and this circumstance does not impact psychosocial health and ART adherence [53]. Greater social acceptance makes it easier for PLWH to live openly with their disease [54]. Studies show that policies addressing stigma against LGBTQ+ people are significant structural determinants of the HIV risk in Europe [46].

Furthermore, it is affirmed that to combat HIV-induced stigma and discrimination, evidence-based interventions are needed to reduce HIV prevalence (Greenwood et al., 2022).

Further research aimed at developing appropriate evidence-based interventions to reduce stigma and discrimination is recommended.

## 5. Limitations

There are several limitations to this study. (1) Due to heterogeneity, not all affected groups with HIV/AIDS could be included. (2) The geographical distribution of the interviewed patients was chosen according to the possibility of field access. (3) Only German-speaking participants and no PLWH with a migration background were included. (4) Another limitation is the self-reported assessment of adherence and quality of life. Due to the small number of respondents, no generally valid statements can be made regarding the QoL. (5) No conclusions can be drawn for non-industrialized countries. (6) There were significantly more men than women because women were more difficult to recruit than men.

## 6. Conclusions

The chronicity of HIV is characterized by the determinants of a very well-tolerated therapy, the integration of HIV into private and professional everyday life without adjustment efforts, and the achievement of a high quality of life. Coping with the diagnosis still has a major part in dealing with the overall situation. Stigmatization and discrimination are still the main aspects that have to be addressed. In addition, addressing structural discrimination and stigma has an impact on prevention and reducing HIV prevalence. Support services in health care facilities should increasingly include these issues in their counseling services.

## Figures and Tables

**Figure 1 ijerph-20-03000-f001:**
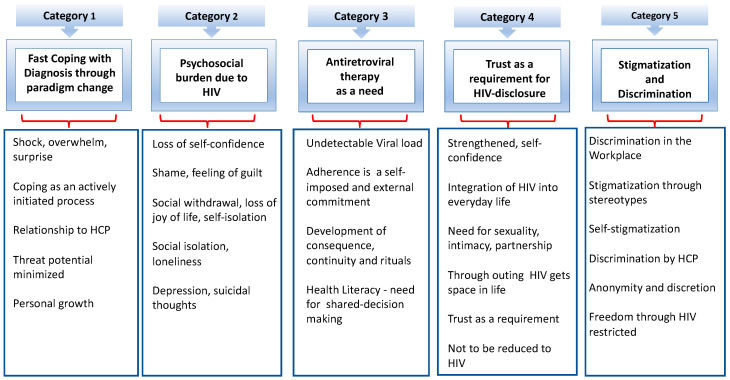
The five core categories (own presentation).

**Figure 2 ijerph-20-03000-f002:**
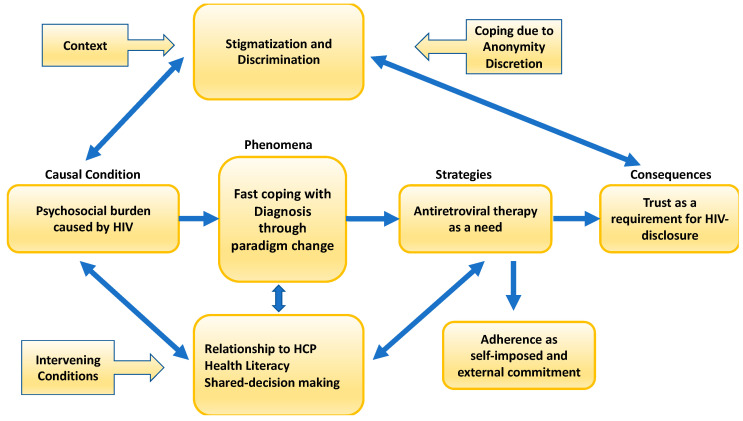
Coding paradigm (own presentation).

**Figure 3 ijerph-20-03000-f003:**
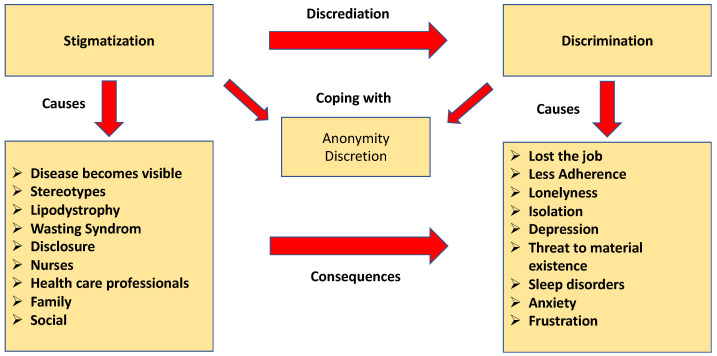
Stigmatization and discrimination (own presentation).

**Table 1 ijerph-20-03000-t001:** Sample qualitative interview agenda questions.

Domain Patient Interview	Questions
HIV/AIDS	How are you doing right now? Tell me about the diagnosis of HIV? How long have you been living with HIV?
Antiretroviral therapy Relationship with the treatment team	Tell me about the antiretroviral therapy. What are the challenges? How can the regular intake be adhered to? Describe the relationship with the treatment team.
Discrimination and stigma	Tell me about experiences and exposure to discrimination and stigma.
Gender-specific characteristics	How does a woman/man live with HIV? Are there gender-specific differences?
Physical and psychosocial burden	What burdens arise in daily life? How is the coping?

**Table 2 ijerph-20-03000-t002:** Demographic and Life Context Description of Women and Men (N = 25).

PLWH	Sex	Sexual Orientation	Age	HIV in Years	ART	Occupation	Marital Status	Country
1	m	MSM	30	6		employed	partnership	A
2	m	MSM	55	20		unemployed	single	A
3	m	MSW	45	25		employed	partnership	A
4	m	MSM	33	4		employed	single	A
5	f	heterosex	55	25		employed	partnership	A
6	m	MSM	22	2		employed	partnership	A
7	m	MSM	50	30		retired	partnership	G
8	m	MSM	25	1.5		employed	single	G
9	m	MSM	42	15		employed	single	A
10	f	heterosex	60	30		unemployed	single	A
11	m	MSM	64	38		retired	single	A
12	f	heterosex	56	36		retired	single	A
13	f	heterosex	49	21		employed	single	A
14	f	heterosex	58	38		employed	partnership	A
15	m	MSM	34	2		employed	partnership	A
16	f	heterosex	52	17		employed	partnership	A
17	m	MSM	25	2		employed	single	A
18	f	heterosex	25	25		employed	partnership	A
19	m	MSM	45	4		employed	partnership	A
20	m	MSM	54	1		employed	single	G
21	f	heterosex	43	21		employed	single	G
22	f	heterosex	56	31		employed	single	G
23	m	MSM	60	30		employed	partnership	G
24	m	MSM	32	2		employed	single	A
25	f	heterosex	50	32		employed	single	G

## Data Availability

Data is not available due to privacy and ethical guidelines.

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
