# Peer review of "People Living with HIV and AIDS: Experiences towards Antiretroviral Therapy, Paradigm Changes, Coping, Stigma, and Discrimination—A Grounded Theory Study"

_ijerph, 2023, doi:10.3390/ijerph20043000_

Round 1

Reviewer 1 Report

1. The first letters of the title words should be capitalized

2.(PLWH)=(PLHIV) line 11

3. Therapy adherence=Medication Adherence line 13

4. Keywords< 5

5. Please  can be refer to Adherence to Medication and Physical Activity among People Living with HIV/AIDS

6. study not studies you had one study line 450

7. Please add conflicting studies to the discussion

8. The way of writing references is not correct. Please write all references in the same style according to the journal guidelines

Author Response

1. The first letters of the title words should be capitalized

Reply: Thank you very much for this comment. The first letters are now capitalized in the title “People Living With…” (Title)

2.(PLWH)=(PLHIV) line 11; 3. Therapy adherence=Medication Adherence line 13

Reply: I used the abbreviation PLWH instead of PLWHIV. Most studies use the neutral and non-stigmatizing abbreviation.

4. Keywords< 5

Reply: I reduced the keywords

5. Please can be refer to Adherence to Medication and Physical Activity among People Living with HIV/AIDS

Reply: The reference to medication adherence in daily life was the subject of the study, not so much physical activity

6. study not studies you had one study line 450

Reply: It is corrected

7. Please add conflicting studies to the discussion

Reply: Another conflicting study is added in the discussion.

“Knight et al., 2019 deny this finding and point out that initiation of ART immediately after diagnosis predominantly refers to countries where universal and unrestricted health care is available for PLWH. Areas of sub-Saharan Africa but also countries in Asia do not have these possibilities.”

8. The way of writing references is not correct. Please write all references in the same style according to the journal guidelines

Reply: During der last publication, the journal did the correction of the references in the main text. Is this still so?

Reviewer 2 Report

The manuscript entitled “People living with HIV and AIDS: Experiences towards an- 2 tiretroviral therapy, paradigm change, cooping, stigma, and dis- 3 crimination. A Grounded Theory Study” authored by Beichler et al., has investigated the effect of HIV infection on the HIV positive people from the society where they live and work. The study also included the positive effect in their life brought by the anti-retroviral therapy. The study was designed in accordance with the proposal of the project and the manuscript is written in standard English language, however there are minor typographic errors which need to be corrected. Abbreviations should be expanded when used for the first time.

Author Response

We thank the reviewer for carefully reading and appreciating our manuscript.

Additionally, we thank the reviewers for their many helpful, constructive suggestions to improve this manuscript.

Thank you very much for this helpful comment.

Comment: Abbreviations should be expanded when used for the first time.

Reply: The whole text has been corrected once again and all typographic errors are hopefully corrected

The whole text for all abbreviations is corrected again

Reviewer 3 Report

The article is obviously interesting. However certain point are unclear and should be corrected before the next step.

1. From the background section is not clear what is a gap in current knowledge in the research question. There is information about stigmatization, but know any introduction about the subjective perspective of PLWH concerning living, 16 coping, and managing HIV/AIDS in daily life.

2. The aim of study is not the same in the main text. In the abstract it is better and clear. Please correct it in the main text.

3. Please add to the subsection "2.1. Recruitment and Participation" the study timeline and location.

Author Response

We thank the reviewer for carefully reading and appreciating our manuscript.

Additionally, we thank the reviewers for their many helpful, constructive suggestions to improve this manuscript.

1. From the background section is not clear what is a gap in current knowledge in the research question. There is information about stigmatization, but know any introduction about the subjective perspective of PLWH concerning living, 16 coping, and managing HIV/AIDS in daily life.

Reply: The background was revised and supplemented with content on coping, subjective view and disease management.“Particularly in highly industrialized countries, HIV can now be integrated and managed particularly well in daily life. Modernization of diagnostics and early initiation of antiretroviral therapy make optimal disease management possible, accompanied by seam-less care in the treatment team (Hoffmann&Rockstroh, 2022).”

2. The aim of study is not the same in the main text. In the abstract it is better and clear. Please correct it in the main text.

Reply: Thank you very much for this helpful comment.We have clarified the aim in the main text:“This grounded theory study aimed to examine the subjective perspective of PLWH concerning living, coping, and managing HIV/AIDS. We also aimed to describe an up-to-date picture of PLWH including their personal views on living with HIV/AIDS in everyday life, at work and in society in Austria and Germany.”

3. Please add to the subsection "2.1. Recruitment and Participation" the study timeline and location.

Reply: We have added the timeline in this section. Interviews were mostly performed via video-call.“People participating in the study were recruited via physicians and through civil society organizations. Participants were made aware of the study via flyers and were directly approached and referred by HCP from January to September 2022.”